# Diversity of Microorganisms in Biocrusts Surrounding Highly Saline Potash Tailing Piles in Germany

**DOI:** 10.3390/microorganisms9040714

**Published:** 2021-03-30

**Authors:** Ekaterina Pushkareva, Veronika Sommer, Israel Barrantes, Ulf Karsten

**Affiliations:** 1Department of Applied Ecology and Phycology, Institute of Biological Sciences, University of Rostock, 18059 Rostock, Germany; veronika.sommer@uni-rostock.de (V.S.); ulf.karsten@uni-rostock.de (U.K.); 2Department of Biology, Botanical Institute, University of Cologne, 50674 Cologne, Germany; 3Research Group Translational Bioinformatics, Institute for Biostatistics and Informatics in Medicine and Ageing Research, Rostock University Medical Center, 18057 Rostock, Germany; israel.barrantes@uni-rostock.de

**Keywords:** biocrusts, diversity, microorganisms, potash mining, high-throughput sequencing

## Abstract

Potash tailing piles located in Germany represent extremely hypersaline locations that negatively affect neighbouring environments and limit the development of higher vegetation. However, biocrusts, as cryptogamic covers, inhabit some of these areas and provide essential ecological functions, but, nevertheless, they remain poorly described. Here, we applied high-throughput sequencing (HTS) and targeted four groups of microorganisms: bacteria, cyanobacteria, fungi and other eukaryotes. The sequencing of the 16S rRNA gene revealed the dominance of Proteobacteria, Cyanobacteria and Actinobacteria. Additionally, we applied yanobacteria-specific primers for a detailed assessment of the cyanobacterial community, which was dominated by members of the filamentous orders Synechococcales and Oscillatoriales. Furthermore, the majority of reads in the studied biocrusts obtained by sequencing of the 18S rRNA gene belonged to eukaryotic microalgae. In addition, sequencing of the internal rDNA transcribed spacer region (ITS) showed the dominance of Ascomycota within the fungal community. Overall, these molecular data provided the first detailed overview of microorganisms associated with biocrusts inhabiting highly saline potash tailing piles and showed the dissimilarities in microbial diversity among the samples.

## 1. Introduction

The production of potash fertilizers results in great amounts of salt waste, which is deposited as potash tailing piles. Potash mainly consists of the by-product NaCl and, therefore, has negative effects on the environment. The impact of salt from the potash tailing piles is reflected in the surrounding ecosystems, which leads to anthropogenic salinization of ground water, freshwater systems and soil [1]. Due to the limited higher plant vegetation in the surroundings of the potash tailing piles, biocrusts appear to play an important role. They consist of several micro-(microalgae, cyanobacteria, bacteria, microfungi) as well as macro-organisms (lichens, mosses) that live in the first millimetres of the soil [2]. Biocrusts often prevail in extreme environments where higher plant establishment is inhibited due to harsh abiotic conditions.

Considering that without a base sealing, large amounts of NaCl from the potash tailing piles leak into the surroundings (soil, ground water etc.), halophytes common in coastal habitats [3,4], as well as salt-tolerant fauna [5,6], are getting established in these anthropogenically disturbed areas. Various studies have addressed the occurrence of microbial phototrophs [7], fungi [8,9,10] and bacteria as well as Archaea [11,12,13,14] in hypersaline environments alone. However, microbial community composition in biocrusts from naturally saline habitats was described in a few studies only [15,16] and that of German potash tailing piles exclusively by Sommer et al. [17,18]. Besides, both latter studies, using traditional culture-dependent methods, showed a diverse community of microbial phototrophs in the biocrusts [17,18].

There are many potash tailing piles in Germany. Some of them are still active and continue to enlarge because of fertilizer mining. Therefore, it is crucial to understand the impact of salt on biocrust microbiota inhabiting the surroundings. Here, we applied high-throughput sequencing (HTS) to obtain thorough and comprehensive insight into biocrust microbial community composition, with an emphasis on bacteria, cyanobacteria, fungi and other eukaryotes in the previously partially described biocrusts [17,18]. These data will help us to better understand the presence and abundance of specific salt-tolerant microorganisms in anthropogenically disturbed mining areas. In addition, such information is essential for any restoration efforts of potash tailing piles by, for example, artificial establishment of biocrusts in order to lower the salt leachate.

## 2. Materials and Methods

A total of twelve highly saline biocrust samples from the areas surrounding five potash tailings piles in Germany (Shreyahn (SY), Oedesse (OD), Wietze (WT), Teutschenthal (TT), the not named site (NN)) were used in this study. Site descriptions and sampling as well as the chemical parameters and microphototrophic community composition were presented in two previous publications [17,18] and are summarised in Appendix A.

Total DNA was extracted from 250 mg biocrust samples using a PowerSoil DNA Isolation Kit (MOBIO, Carlsbad, CA, USA) according to the manufacturer’s instructions and DNA concentrations were quantified using a Qubit 3.0 Fluorometer. Furthermore, the DNA was sent to Microsynth AG (Balgach, Switzerland), where PCR and sequencing using the Illumina MiSeq platform (v3, 2 × 300 bp) were performed. Four groups of microorganisms were targeted in the amplification step: bacteria (V3–V4 region of the 16S rRNA gene), cyanobacteria (V4 region of the 16S rRNA gene), fungi (rDNA internal transcribed spacer region (ITS)) and other eukaryotes (18S rRNA gene). The primers used in this study and their descriptions are presented in Appendix A. The raw sequencing data were deposited in the European Nucleotide Archive (ENA) under the accession number PRJEB43759.

Quality control and assembly of the reads were performed using the PANDAseq program (version 2.11 [19]). Operational taxonomic units (OTUs) were obtained using USEARCH (version 6.1.544 [20]) from the *pick_open_reference_otus.py* script of QIIME 1.9.1 [21] and later assigned to the appropriate taxa using the following databases with 97% identity threshold: 16S rRNA Greengenes (version 13.8 [22]) for the bacteria and cyanobacteria; UNITE (version 12_11 [23]) for the fungi and SILVA (version 132 [24]) for the other eukaryotes. Several samples resulted in the absence or a low number of reads and thus were excluded from any further analyses (Appendix A). Eventually, the highly saline conditions of biocrusts seemed to interfere with DNA extraction or even further with PCR, as it was challenging to isolate pure algal cultures from the studied samples in the previous study [18].

The statistical analyses and figures were performed in RStudio (version 1.4.1106). Alpha diversity indices (Shannon and Simpson) were calculated using the package *vegan* [25]. The community dissimilarities between the studied biocrust samples were evaluated using a similarity profile routine (SIMPROF [26]). Pearson correlation coefficients were calculated to assess the relationship between alpha diversity indices and soil parameters such as pH, total organic carbon (TOC), total organic nitrogen (TON) and the C/N ratio.

## 3. Results and Discussion

High-throughput sequencing produced 143,936 bacterial, 22,655 cyanobacterial, 407,178 eukaryotic and 155,349 fungal reads (1527, 87, 329 and 189 OTUs, respectively). These numbers were much lower than those observed in other biocrusts [27,28,29]. The low number of reads, together with several failed samples, could suggest that highly saline conditions might either inhibit biocrust microbiota or negatively influence DNA extraction.

The alpha diversity indices revealed higher bacterial diversity across the studied biocrusts than other groups of microorganisms (Figure 1). In general, the bacterial community in the biocrusts surrounding potash tailing piles was dominated by Proteobacteria (29% of total bacterial reads, 446 OTUs), Cyanobacteria (16% of total bacterial reads, 87 OTUs), Actinobacteria (15% of total bacterial reads, 255 OTUs), Planctomycetes (14% of total bacterial reads, 227 OTUs), Bacteroidetes (10% of total bacterial reads, 181 OTUs) and Chloroflexi (8% of total bacterial reads, 124 OTUs) (Figure 2a). Overall, the distribution of bacterial orders in the studied biocrusts was similar to those of other biocrusts around the world [27,30,31,32]. Besides, representatives of Bacteroidetes and Gammaproteobacteria were shown to be salt-tolerant in soils [33]. In addition, genera such as *Marinobacter* (19 OTUs), *Halomonas* (three OTUs) and *Halorhodospira* (one OTU) identified within the Gammaproteobacteria as well as the genus *Salinimicrobium* (two OTUs) from the Bacterioidetes are common for saline to extremely saline environments [34,35,36]. However, records of these genera in biocrusts are very rare, which underlines the unique conditions of the studied sampling sites.

The high numbers of cyanobacteria obtained with general bacterial primers in the studied biocrusts confirmed their importance in such extreme environments (Figure 2a). Moreover, a detailed investigation of cyanobacteria with cyanobacteria-specific primers (Figure 2b) showed the dominance of the filamentous orders Synechococcales (64% of total cyanobacterial reads, 54 OTUs) and Oscillatoriales (31% of total cyanobacterial reads, 21 OTUs). The dominance of filamentous cyanobacteria from these two orders has been previously observed in biocrusts from other extreme environments such as, for example, Arctic soils [37,38,39]. In addition, one OTU recorded at two sites (NN and TT) was identified as the salt-tolerant genus *Halomicronema* from the order Synecchococcales [40].

In contrast to the filamentous cyanobacteria, members of the heterocystous order Nostocales, which are able to fix atmospheric nitrogen (N_2_), exhibited much lower abundance (3% of total cyanobacterial reads, six OTUs) than in biocrusts from other parts of Germany [37]. This could indicate that the biocrusts in potash tailing piles are poorly-developed and that ecological processes induced by these N-fixing microorganisms have slowed down [41]. Likewise, members of the order Chroococcales usually constitute a minor fraction within biocrust cyanobacteria [28,37] and only a small proportion of this order was detected in the biocrusts surrounding potash tailing piles (1% of total cyanobacterial reads, three OTUs).

The eukaryotic community in the studied biocrusts was dominated by the clade Archaeplastida (59% of total eukaryotic reads, 95 OTUs; Figure 2c). The phylum Chlorophyta, with the dominance of the two families (Chlorophyceae and Trebouxiophyceae), was the only representative of the clade. These eukaryotic microalgae are typically found in biocrusts [42] and were also reported in the studied potash tailing piles [17,18]. They produce organic osmolytes, which serve as salt protectants [43], and extracellular polymeric substances (EPS) that provide several functions including drought tolerance [44], which, in turn, might have a salt-buffering effect. In addition, the abundance of Archaeplastida increased corresponding to the age of the potash piles. For example, this clade in the two old potash piles (OD and SY) constituted 99.9% of whole eukaryotic community, while in the young piles (NN and TT), it was only 43.5%. Furthermore, the majority of the SAR (Stramenopila, Alveolata and Rhizaria) clade (28% of total eukaryotic reads, 134 OTUs) belonged to the phylum Ochrophyta, which includes diatoms, yellow-green and golden algae. Besides, 17% of total eukaryotic reads corresponded to one OTU assigned to the class Xanthophyceae, which resembled the low number of morphologically identified Xanthophyceae in biocrusts from the potash tailing piles’ surroundings [17].

Sequencing of the ITS region (Figure 2d) showed that Ascomycota was the prevalent fungal phylum, constituting 78% of the total fungal reads (106 OTUs). These fungi, similar to other halotolerant microorganisms, produce exopolysaccharides [45], which might explain their dominance in the highly saline environment. Likewise, Basidiomycota is a potentially halotolerant fungal phylum [45] and a high number of reads was recorded in one sampling site (WT-4). In addition, a large part of the OTUs (21% of total fungal reads) was not assigned to any fungal phyla, either due to database incompleteness or to the presence of new undescribed species.

Multivariate analysis such as SIMPROF based on the amplicon dataset showed that the biocrust samples clustered irrespectively of the origin site, indicating the heterogeneity of microorganisms within individual saline potash tailing piles (Appendix A). In addition, soil chemical parameters such as pH, C/N ratio and contents of TOC and TON are well known to support biocrust microbiota [2]. Here, we found no significant correlations between alpha diversity indices and these soil chemical parameters [17], suggesting that other abiotic factors might shape microbial communities in highly saline biocrusts.

## 4. Conclusions

To the best of our knowledge, this study is the first in-depth overview of the microbiota inhabiting highly saline potash tailings piles located in Germany using HTS. Furthermore, this work is complementary to the previous studies in this area and could be used as a baseline for future research aiming to restore potash tailings piles with biocrusts as primary microbiotic vegetation.

## Figures and Tables

**Figure 1 microorganisms-09-00714-f001:**
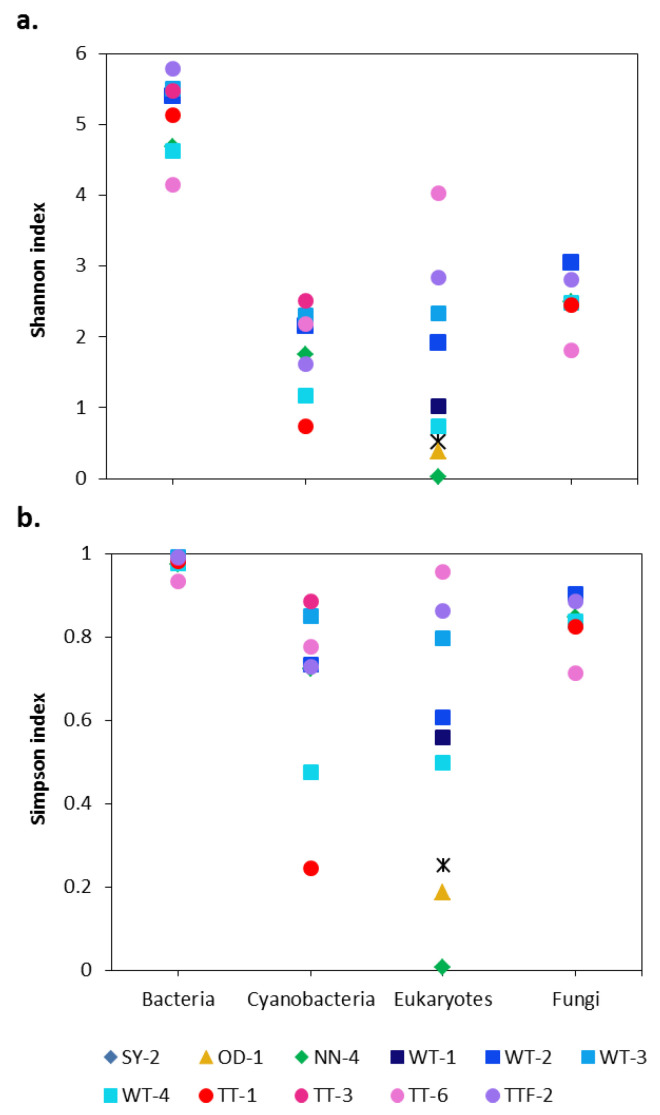
Alpha diversity indices in the studied biocrusts: (**a**) Shannon and (**b**) Simpson. SY, OD, WT, TT and NN correspond to Shreyahn, Oedesse, Wietze, Teutschenthal and the not named site, respectively.

**Figure 2 microorganisms-09-00714-f002:**
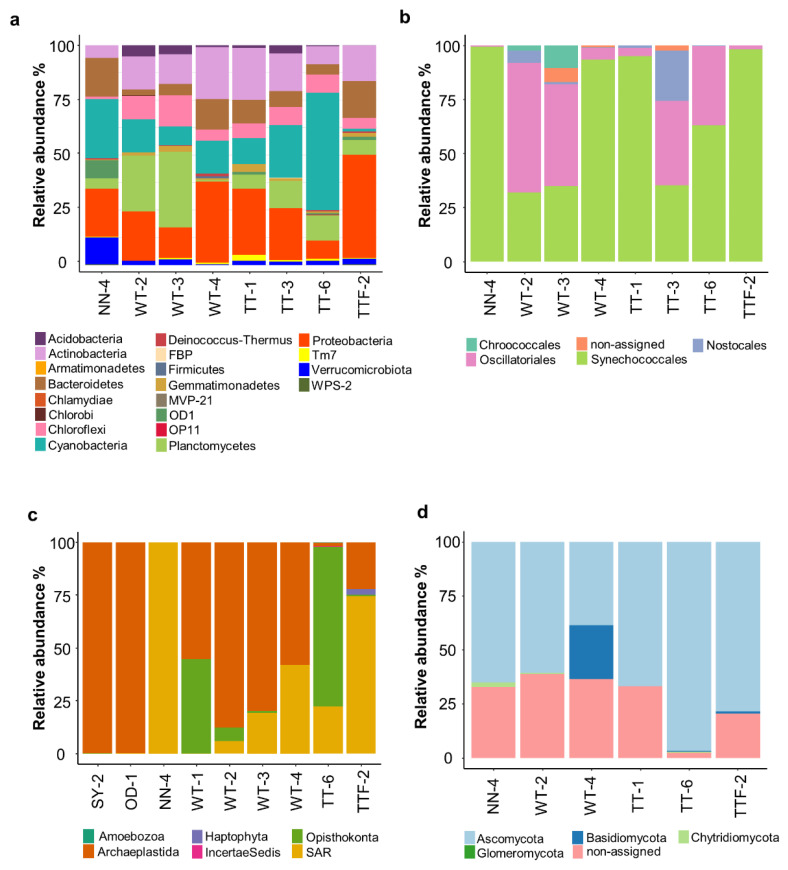
Relative abundance of bacteria (**a**), cyanobacteria (**b**), eukaryotes (**c**) and fungi (**d**) in the studied biocrusts. SY, OD, WT, TT and NN correspond to Shreyahn, Oedesse, Wietze, Teutschenthal and the not named site, respectively.

## Data Availability

The raw sequencing data are available at the European Nucleotide Archive (ENA) under the accession number PRJEB43759.

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
