# Peer review of "Diversity of Microorganisms in Biocrusts Surrounding Highly Saline Potash Tailing Piles in Germany"

_microorganisms, 2021, doi:10.3390/microorganisms9040714_

Round 1
Reviewer 1 Report
Major Comments
I would highly recommend that the authors consider re-analyzing their sequencing data with Qiime2, using the DADA2 plugin. This method supports all paired end raw amplicon sequencing data from Illumina Miseq and provides exact amplicon sequence variants yielding higher taxonomic resolution than OTUs. As it stands, resolution to the order or family level makes most functional inference of taxa irrelevant. The authors would also be able to compare sequence variants across sampling sites and make informed conclusions about presence and diversity of microorganisms across sites.
Callahan, B., McMurdie, P. & Holmes, S. Exact sequence variants should replace operational taxonomic units in marker-gene data analysis. ISME J 11, 2639–2643 (2017). https://doi.org/10.1038/ismej.2017.119
Additionally, as cyanobacteria seem to be an important component of highly saline biocrust communities, I would also recommend the use of Cydrasil (cydrasil.org), a publicly available, manually curated, cyanobacterial reference package, for cyanobacterial taxonomic placements. This package contains over 1,400 cyanobacterial representatives from a wide range of habitats and should allow you to achieve higher taxonomic resolution, in most cases, down to the genus level. This higher level of taxonomic resolution would make descriptions of the cyanobacterial component of these communities much more informative, comparable and relevant to others studying biocrust microbial ecology.
Lastly, I am unsure of the exact purpose of this manuscript as it is either to compare the microbial diversity across a variety highly saline potash tailings piles (no statistical analyses were done) or merely to report the components of biocrusts at these locations using 12 samples across 5 sites. The problem is that no comparative statistical analyses were performed, nor does the sample size per site does seem large enough to be representative of each location (n<3 in most cases). The highly saline potash pile tailings represent a unique environment for biocrust to develop and understanding the microbial ecology of these biocrust communities could lead to beneficial applications, as such the authors should attempt at least basic interrogation of the sequencing data. There are simple questions that can be addressed and statistically supported in order to make a statement about these communities, such as:
-Is the level of alpha diversity *significantly* similar or different across all sampled sites and samples?
-Do all potash piles have similar microbial components? How similar or different are the microbial communities across sites (PERMANOVA/PCoA/NMDS)?
-Which exact amplicon sequence variants (see above) occur across all sampled sites, if any?
-Which exact amplicon sequence variants drive the overall differences in community across sites (SIMPER)?
Minor Comments
Line 60: What was the mass of biocrust used for DNA extraction?
Line 63: What chemistry was used on the Illumina MiSeq platform? 2 x 250 paired end?
Lines 65-72: Are these new primers sets? How were they generated? If not new, the primers sets should be appropriately referenced.
Lines 84-87: Number of reads in high-throughput sequencing are not an accurate metric of microbial biomass. Number of reads/counts can be influenced by extractions conditions of a kit, mass of biocrust extracted, sequencer chemistry, etc.
Lines 125-127: This statement needs a reference.
The authors state that 12 biocrust samples from 5 different locations were sampled and sequenced (Lines 55-56), but then go on to say that several samples were excluded from further analysis due to absent or low reads (Lines 79-80). The authors do not state which samples were excluded from which analysis. This leads to confusion in Figure 2., where sample number changes in each relative abundance graph. For example, Figure 2a shows 8 samples, but Figure 2c shows 9 samples. These discrepancies must be explained in the methods or figure captions.
Throughout the manuscript the authors suggest that microbial components of these highly saline biocrust use exopolymeric substances to tolerate highly saline conditions (Lines 97-98, 119-120, 134-137, 149-151). While this may be true, making functional inferences on phylum level taxonomic resolution is pointless, the production of exopolymeric substances is a ubiquitous trait across soil microorganisms.
Reviewer 2 Report
Dear authors and the editor,
Here is the review of the manuscript entitled "Diversity of microorganisms in biocrusts surrounding highly saline potash tailings piles in Germany " written by Ekaterina Pushkareva and the co-authors. The aim of the paper is to analyze composition of microbial communities in surroundings of hypersaline highly saline potash tailings piles in Germany. Groups of organisms that were studied are Bacteria, cyanobacteria, fungi and other eukaryotes. High-throughput sequencing (HTS) of the 16S rRNA using Illumina MiSeq platform revealed that the dominant taxa in hypersaline habitat are Proteobacteria, Cyanobacteria and Actinobacteria. Application of cyanobacteria specific primers showed that orders Synechococcales and Oscillatoriales are dominant. HTS sequencing of the fungal ITS region showed that the phylum Ascomycota was the most dominant within the fungal community.
In my opinion the study has a potential to be accepted for publication in Microorganisms journal but it needs to be thoroughly revised.
The major problem is that statistical methods used were not adequately described (diversity indices, relative abundance etc.). Obtained results were not thoroughly commented and the obtained results from different sites were not mutually compared and statistically analysed. The English language used in the manuscript needs moderate revision. There is no supplementary material (e.g. OTU table with taxonomic identifications).
Suggested changes/additions and my questions are included in the pdf of the manuscript file attached.
Best wishes,
Reviewer

Round 2
Reviewer 1 Report
The authors have made a majority of the requested changes throughout the manuscript and I find that the manuscript is overall improved.
Below are my final comments:
I was pleased to see the authors made some effort to analyze the microbial compositions of the different potash tailings pile communities and come to some conclusion about their similarity/dissimilarity. The conclusion that all sampled sites were highly dissimilar from each other implies further studies into restoration of potash tailings piles will perhaps be complicated due to each site being highly influenced by local conditions. Interpretations such as these are the difference between publishing data and publishing discoveries.
The authors decision to stay with their current bioinformatic methods (Qiime1/OTUs) is sufficient for a first look at microbial communities in a new environment. I would like to stress that my suggestions to update bioinformatic methods and use supplemental taxonomical tools was an attempt to extract more information out of the provided data. The benefits of using Qiime2/DADA2 to acquire ASVs is self-evident in current literature. Cydrasil has been used in many published peer-reviewed manuscripts, including journals such as Microorganisms, The ISME Journal, Frontiers in Microbiology and is gaining traction in the study of cyanobacteria in many microbial systems. My advice for future studies would be not to disregard tools suggested by reviewers merely because they are new and not cited over 1,500 times.
My last recommendation would be to add a sentence to the abstract stating the conclusions made about the dissimilarity in microbial diversity between the different sampled sites. With this final change, I have no further objections to the publication of this manuscript.
Author Response
Dear reviewer,
Thank you very much for your positive feedback and advice regarding bioinformatic analyses. We will consider them in our future studies.
As suggested, the sentence about the dissimilarity in microbial diversity between the different sampled sites was added to the abstract.
Reviewer 2 Report
Dear authors and editor,
After revision, I find the manuscript suitable for publication in Microorganisms.
Best
Author Response
Dear reviewer,
Thank you very much for your positive feedback and accepting the manuscript.